# Health service readiness to provide care for HIV and cardiovascular disease risk factors in low- and middle-income countries

**Neil Cockburn**[1], **David Flood**[2], **Jacqueline A. Seiglie**[3], **Jennifer Manne-Goehler**[4], **Krishna Aryal**[5], **Khem Karki**[6], **Albertino Damasceno**[7], **Rifat Atun**[8], **Sebastian Vollmer**[9], **Till Bärnighausen**[10], **Pascal Geldsetzer**[11,12], **Mary Mayige**[13], **Lisa Hirschhorn**[14], **Justine Davies**[1] *

1 Institute of Applied Health Research, University of Birmingham, Birmingham, United Kingdom, 2 Department of Medicine, Division of Hospital Medicine, University of Michigan, Ann Arbor, Michigan, United States of America, 3 Department of Medicine, Diabetes Unit, Massachusetts General Hospital, Harvard Medical School, Boston, Massachusetts, United States of America, 4 Division of Infectious Diseases, Brigham and Women's Hospital, Harvard Medical School, Boston, Massachusetts, United States of America, 5 Public Health Development Organization, Kathmandu, Nepal, 6 Department of Community Medicine, Maharajganj Medical College, Institute of Medicine, Kathmandu, Nepal, 7 Faculty of Medicine, Eduardo Mondlane University, Maputo, Mozambique, 8 Department of Global Health & Population, Department of Health Policy & Management, Harvard T.H. Chan School of Public Health, Boston, Massachusetts, United States of America, 9 Center for Modern Indian Studies, University of Göttingen, Göttingen, Germany, 10 Heidelberg Institute of Global Health, University of Heidelberg, Heidelberg, Germany, 11 Department of Medicine, Division of Primary Care and Population Health, Stanford University, Stanford, California, United States of America, 12 Chan Zuckerberg Biohub, San Francisco, California, United States of America, 13 National Institute for Medical Research, Dar es Salaam, Tanzania, 14 Ryan Family Center on Global Primary Care, Feinberg School of Medicine, Northwestern University, Chicago, Ilinois, United States of America

* j.davies.6@bham.ac.uk

**Data Availability Statement:** Data are held by the Demographic and Health Survey and are available to download with permission from https://

## Abstract

Cardiovascular disease risk factors (CVDRF), in particular diabetes and hypertension, are chronic conditions which carry a substantial disease burden in Low- and Middle-Income Countries. Unlike HIV, they were neglected in the Millenium Development Goals along with the health services required to manage them. To inform the level of health service readiness that could be achieved with increased attention, we compared readiness for CVDRF with that for HIV. Using data from national Service Provision Assessments, we describe facility-reported readiness to provide services for CVDRF and HIV, and derive a facility readiness score of observed essential components to manage them. We compared HIV vs CVDRF coverage scores by country, rural or urban location, and facility type, and by whether or not facilities reported readiness to provide care. We assessed the factors associated with coverage scores for CVDRF and HIV in a multivariable analysis. In our results, we include 7522 facilities in 8 countries; 86% of all facilities reported readiness to provide services for CVDRF, ranging from 77–98% in individual countries. For HIV, 30% reported of facilities readiness to provide services, ranging from 3–63%. Median derived facility readiness score for CVDRF was 0.28 (IQR 0.16–0.50), and for HIV was 0.43 (0.32–0.60). Among facilities which reported readiness, this rose to 0.34 (IQR 0.18–0.52) for CVD and 0.68 (0.56–0.76) for HIV. Derived readiness scores were generally significantly lower for CVDRF than for

dhsprogram.com/data/Using-Datasets-for-Analysis.cfm.

**Funding:** The authors received no specific funding for this work.

**Competing interests:** The authors have declared that no competing interests exist.

HIV, except in private facilities. In multivariable analysis, odds of a higher readiness score in both CVDRF or HIV care were higher in urban vs rural and secondary vs primary care; facilities with higher CVDRF scores were significantly associated with higher HIV scores. Derived readiness scores for HIV are higher than for CVDRF, and coverage for CVDRF is significantly higher in facilities with higher HIV readiness scores. This suggests possible benefits from leveraging HIV services to provide care for CVDRF, but poor coverage in rural and primary care facilities threatens Sustainable Development Goal 3.8 to provide high quality universal healthcare for all.

## Introduction

Globally, nearly 18 million premature deaths in 2019 were due to cardiovascular diseases (CVD). Of these, 75% were in low- or middle-income countries (LMICs), where they are leading causes of death and Disability Adjusted Life Years (DALYs) in adults [1]. Diabetes and hypertension are key risk factors for CVD and amongst the top three risk factors for deaths and disability, globally [2].

Starting in 2015, the Sustainable Development Goals (SDGs) included goal 3.4 aiming to reduce by 1/3 premature mortality due to non-communicable diseases [3]. This requires managing CVD, and, importantly, managing cardiovascular disease risk factors (CVDRF) for the primary prevention of CVD. However, cascades of care (which assess whether patients with disease are diagnosed, treated, or controlled) indicate poor health system performance for CVDRFs; individual patient data from multiple LMICs show large drop offs at all cascade stages for both diabetes and hypertension [4–7]. Studies have also shown that meeting targets for improving the cascade of care would be both effective and cost effective, with incremental cost effectiveness ratios below the WHO thresholds of 3 times GDP per capita per DALY averted [8,9]. Strong health systems with sustained investment in CVDRF management are crucial to meet SDG 3.4, and, in particular, integration into primary healthcare. However, evidence shows health-system strengthening necessary to ensure that services are prepared to manage CVDRF is insufficient [10–12].

In contrast to CVDRF, which gained global attention as the SDGs started in 2015, HIV has received international attention as a chronic infectious disease for a longer period of time and especially since the 2000s, through the Millenium Development Goals (MDG) [13]. The global attention and funding afforded during the MDG era was associated with significant achievements in improving access to healthcare for HIV [13]. Latterly HIV has continued to receive attention thorough the UN's 95:95:95 targets to detect 95% of people with HIV, to treat 95% of people who have HIV detected, and to supress viral load in 95% of people who are treated, by 2025 [14].

Some have suggested that the investment in health care services needed to deliver these targets for HIV could be leveraged for CVDRF care. Indeed, HIV services integrated into other health services have shown better outcomes for both HIV and other conditions [15]. Successful antiretroviral therapy (ART) programs will treat patients over many years, and the infrastructure to provide this care could be used for other chronic disease care such as CVDRFs [16]. ART is also associated with increased risks of CVD [17], and HIV patients are likely to benefit from integrated CVDRF services [18].

A pivotal factor in ensuring patients get the care they require is the readiness of healthcare facilities to deliver that care. This readiness is formed from the availability of equipment, staff,

information and medicines to manage diseases, as described in the WHO Health System Building Blocks [19]. Two instruments used to capture the readiness of health facilities internationally are the Service Availability and Readiness Assessment (SARA) produced by the World Health Organisation [20], and the Service Provision Assessment (SPA), developed by the USAID Demographic and Health Surveys Program. SPA is based on SARA with some additional questions [21,22]. Whilst these have been used to inform service development at a national level [23–26], they have been underutilised for comparing health service readiness to provide care for CVDRF across countries or with other diseases. In making such comparison, it is possible to start to understand acheivements that can be made and transfer learning between countries.

In this study, we aimed to determine readiness of health services to provide care for CVDRF across multiple countries. To indicate the level of readiness for CVDRF that could be achieved with sustained global attention and investment, we compared CVDRF readiness with that of HIV. To explore the potential for leveraging HIV service readiness to provide care for CVDRF, we also examined the association between CVDRF and HIV readiness.

## Methods

### Study design and data sources

We did a secondary analysis of cross-sectional data from SPA surveys conducted after 2012. Where there were multiple surveys done since 2012 in any one country, we used the most recent survey. These surveys are implemented and analysed by ministries of health to guide national policies.

The SPA questionnaire was first conducted in 1997 and 30 surveys have been conducted in 17 countries in total. Since the survey instrument underwent restructuring in 2012, 16 surveys have taken place in 8 different countries: Afghanistan, Bangladesh, Democratic Republic of Congo (DRC), Haiti, Malawi, Nepal, Senegal and Tanzania.

Surveys are either done in every health facility in a country as a census, or on a representative sample of facilities [20]. Implementation of the SPA instrument is designed to have minor variations from country to country, such as by facility types, or by country-specific medication guidelines (e.g. variation in recommended antihypertensive medications) but the core instrument indicators are the same across countries. Questions are administered to facility staff by trained data collectors who also observe availability of readiness components, e.g., if medicines are reported present, the interviewer will ask to verify their presence. 10% of all facilities sampled are recommended to receive a second visit to validate results. In our analysis, facilities that did not complete the full survey were excluded.

In addition to collecting data on components related to the WHO building blocks, which allows objective observation of readiness to provide care, the surveys also directly ask self-reported questions of facilities' readiness to provide care for specific diseases or disease areas. We refer to observations of components of care such as the presence of medicines as derived readiness which are used to calculate facility readiness scores, and reports of services being provided as facility-reported readiness.

### Outcomes

We report 2 outcomes for all included facilities: the facility-reported readiness to provide care and the derived readiness scores for both CVDRF and HIV. We additionally report derived readiness scores in those facilities reporting readiness only.

## Data definitions and covariates

A full data dictionary is provided in S1 Appendix.

Outcomes are reported by country and, where feasible, disaggregated by facility level (e.g, primary vs secondary or above facility), by public or private management, and by rural or urban location.

Facilities were considered to have facility-reported readiness to provide services for CVDRF care if they responded positively to a general question asking about the provision of services as follows: "Does this facility offer any of the following client services? In other words, is there any location in this facility where clients can receive any of the following services...Diagnosis or management of non-communicable diseases, specifically diabetes, cardiovascular diseases, and chronic respiratory conditions in adults."

For HIV, facilities were considered to have reported readiness to provide services if they responded positively to a question about the provision of all of diagnostic, treatment, or associated management services for HIV. The question was asked as follows: "Does this facility offer any of the following client services? In other words, is there any location in this facility where clients can receive any of the following services...HIV testing and counselling services; HIV/AIDS antiretroviral prescription or antiretroviral treatment follow-up services; or HIV/AIDS care and support services, including treatment of opportunistic infections and provision of palliative care". Preventing Mother-to-Child Transmission services were not considered in this study.

A facility score for CVDRF care (based on observed readiness) was derived for all facilities based upon the presence or absence of up to 19 components required for CVDRF care as documented in the CVD and Diabetes sections of the SARA manual. The SARA manual was used given there is no score for CVDRF in the SPA manual. Nevertheless, all SARA HIV and CVDRF components are in SPA, so the SARA manual can readily be used to derive readiness scores from SPA data. This score was derived as the total number of care components for CVDRF that are present in a facility divided by the total if all components were present.

However, the SARA manual does not contain some CVDRF care-components considered essential in the World Health Organisation's Package of Essential Non-Communicable Disease Intervention (PEN) guidelines, e.g: presence of statins in diabetes management. Therefore, we derived a second CVDRF score based upon the practice recommendations contained in PEN [27]. We used this score in a sensitivity analysis and S2 Appendix contains a side-by-side comparison of components used to create readiness scores.

For HIV care, the observed readiness score was constructed based upon the presence or absence of 19 required care components for HIV care as documented in the SARA manual. The score was derived as the total number of care components for HIV that are present divided by the maximum total if all components were present.

## Ethical statement

This was a secondary analysis of existing anonymised data and no ethical permission was required for the analysis.

## Patient and public involvement

This study did not involve patients or public in its design, conduct or dissemination plans.

## Analyses

All data analysis was conducted using Python 3.8, using the pandas, numpy, seaborn, and statsmodels packages [28–31].

Scores were created for the six domains of health service readiness described in the SARA manual [20], categorised as amenities, equipment, diagnostics, information, medicines, and staffing, although CVDRF scores contain no amenity domain and HIV scores contain no equipment domain. Facility readiness scores were created by averaging scores across each domain, weighted so that each contributed 20% to the overall score to prevent domains with many components from dominating scores. For example, medicines constitute half of CVDRF components. Given that the domain of amenities was shared between CVDRF and HIV, and our intention was to compare coverage scores for HIV with CVRDF, this was not used in deriving the facility coverage scores, apart from audio-visual privacy which was considered in SARA to be an essential amenity for HIV counselling [20]. We also conducted an unweighted sensitivity analysis. Coverage score results are not derived for HIV in Bangladesh as the 2017 survey excluded most HIV items.

All variables are described as frequency (%) when categorical or median (IQR) when continuous. Availability of components of facility coverage scores are displayed using a heat map.

Comparisons of the SARA and PEN methods of deriving readiness scores for CVDRF are shown using mean-Tukey plots and an Intra-Class Correlation was used to compare the two ratings.

Univariable comparisons between CVDRF and HIV scores were done using the Mann-Whitney U test. Associations with CVDRF and HIV coverage scores for all facilities were also assessed using multivariable beta regression analyses in all countries controlling for facility level, funding, and urban or rural location, with country included as a fixed effects variable with differing dispersion parameters. Beta regression uses the beta function link to a generalised linear model and is appropriate to estimating values between 0 and 1 [32]. Bangladesh was excluded from regression modelling due to missing HIV items, and Nepal was excluded due to not collecting data on urban vs rural location of facilities. Similar analyses were done considering each country separately. These analyses were done including all facilities regardless of stated readiness to provide care for CVDRF or HIV. Missing data in SPA surveys are primarily due to skip logic in the design of the survey rather than missing observations [33], and the surveys have been shown to have very low rates of missing data (less than <1%) [23].

## Results

Eight countries (2 low and 6 lower-middle income) representing a total of 13543 facilities had data available from after 2012 (Fig 1 and Table 1). After including only the most recent survey per country, 7911 facilities remained, of which 7606 (96.1%) had completed the survey. 84 surveys took place in Case de santé facilities in Senegal; these surveys were abridged and were excluded from the analysis. Bangladesh did not collect data for 10 of 19 HIV components, and so results for HIV components have been excluded for the 1524 facilities in Bangladesh. For the 305 facilities that did not complete the survey in any country, the reasons provided were facility closure (35%), refusal to participate (23%), respondent unavailable (8%) and other (34%).

Overall, 74.9% of facilities that completed the survey provided primary care, 64.1% of all facilities were publicly funded, and 65.3% of facilities were rurally located.

Results from mean-Tukey plots show only minor differences between SARA and PEN based CVDRF readiness scores (S3 Appendix). We therefore present the results using SARA based scores only in the rest of this manuscript.

Of all facilities, 86.4% reported readiness to provide services for CVDRF and 30.4% for HIV (Table 2). This varied by country, with Haiti having the largest proportion of facilities reporting readiness for CVDRF (97.8%) and Malawi for HIV (62.9%). In general, more secondary care facilities reported readiness for CVDRF (96.9%) or HIV (58.1%) than primary care;

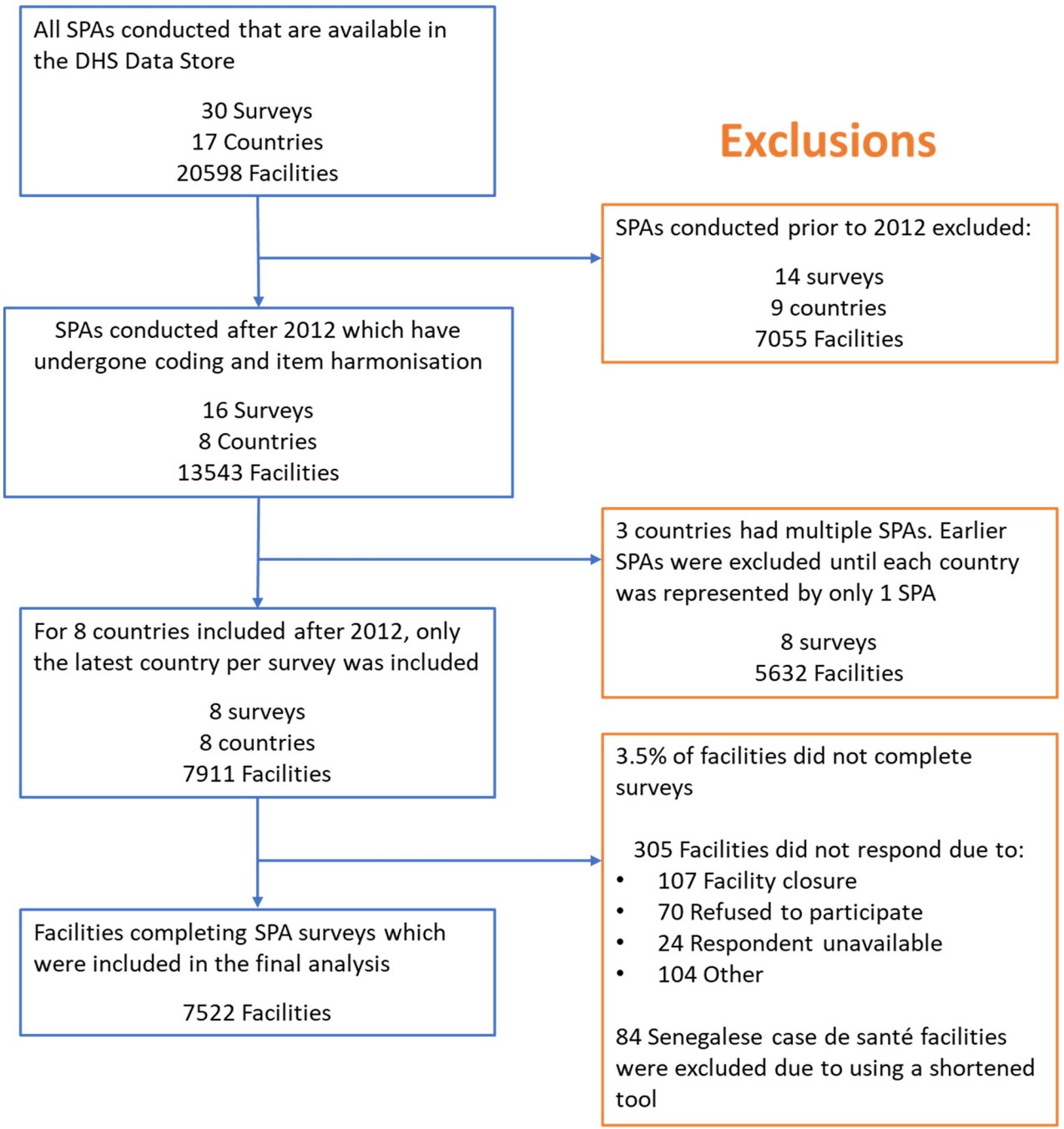

**Fig 1. Facilities included in the analysis.** SPA–Service Provision Assessment; DHS–Demographic and Health Surveys.

(82.9% for CVDRF and 21.2%) for HIV. Facility-reported readiness for CVDRF was greater in private (90.5%) vs public facilities (84.1%) and in urban (90.9%)) vs rural (3550 (82.9%)) areas. While for HIV, the findings were similar (private: 809 [29.9%] vs public: 1481 [30.7%; urban: 760 [33.4%] vs rural: 1469 [34.3%]).

Median (IQR) derived readiness scores for CVDRF and HIV care in all facilities, regardless of whether the facility-reported service provision are presented in Table 2. Considering all

**Table 1. Characteristics of facilities included in the study.**

| | | Overall | Afghanistan* | Bangladesh | DRC | Haiti | Malawi | Nepal*** | Senegal** | Tanzania |
|---|---|---|---|---|---|---|---|---|---|---|
| **World bank Income Group** | | | Low | Lower–Middle | Lower–Middle | Lower–Middle | Low | Lower–Middle | Lower–Middle | Lower–Middle |
| **Year** | | | **2018** | **2017** | **2017** | **2017** | **2013** | **2015** | **2019** | **2014** |
| n | **Total** | 7522 | 142 | 1524 | 1380 | 1007 | 977 | 963 | 341 | 1188 |
| Level of Care, n (% of total surveyed) | **Primary Care** | 5634 (74.9) | 48 (33.8) | 1351 (88.6) | 540 (39.1) | 876 (87.0) | 861 (88.1) | 716 (74.4) | 310 (90.9) | 932 (78.5) |
| | **Secondary Care or above** | 1888 (25.1) | 94 (66.2) | 173 (11.4) | 840 (60.9) | 131 (13.0) | 116 (11.9) | 247 (25.6) | 31 (9.1) | 256 (21.5) |
| Management, n (% of total surveyed) | **Private** | 2703 (35.9) | 108 (76.1) | 212 (13.9) | 548 (39.7) | 663 (65.8) | 499 (51.1) | 192 (19.9) | 73 (21.4) | 408 (34.3) |
| | **Public** | 4819 (64.1) | 34 (23.9) | 1312 (86.1) | 832 (60.3) | 344 (34.2) | 478 (48.9) | 771 (80.1) | 268 (78.6) | 780 (65.7) |
| Geographic Location, n (% of total surveyed) | **Rural** | 4281 (65.3) | 1 (0.7) | 1141 (74.9) | 1076 (78.0) | 629 (62.5) | 678 (69.4) | NA | 0 (0.0) | 756 (63.6) |
| | **Urban** | 2278 (34.7) | 141 (99.3) | 383 (25.1) | 304 (22.0) | 378 (37.5) | 299 (30.6) | NA | 341 (100.0) | 432 (36.4) |

DRC–Democratic Republic of Congo. Surveys are either a complete census or a sample selected to be nationally representative from a complete list.

*Afghanistan is not a nationally representative but a targeted survey of private primary care clinics and all secondary care [25].

**Surveys in Senegal form part of a continuous facility assessment programme where a smaller sample of facilities is surveyed on a yearly basis; only the 2019 wave is considered here [34].

***Nepal did not capture information on urban or rural settings of facilities [35].

countries, median CVDRF derived readiness score (0.28 [IQR 0.16,0.50]) was significantly lower than that for HIV (0.43 [IQR 0.32,0.60]) (p<0.001). CVDRF derived readiness score was significantly lower than HIV score for the Democratic Republic of the Congo (CVDRF, 0.36 [IQR 0.16,0.52] vs HIV, 0.52 [IQR 0.32,0.68], p<0.001), Malawi (CVDRF 0.24 [0.16,0.0.42] vs HIV 0.53 [0.36,0.68], p<0.001), Nepal (CVDRF 0.16 [0.12,0.25] vs HIV 0.32 [0.28,0.39], p<0.001), and Tanzania (CVDRF 0.38 [0.18,0.60] vs HIV 0.52 [0.36,0.69], p<0.001), whereas it was significantly higher for Afghanistan (CVDRF score of 0.52 [IQR 0.44,0.65] vs HIV, 0.40 [0.32,0.47], p<0.001), Haiti (CVDRF 0.42 [0.24,0.58] vs HIV 0.32 [0.28,0.44], p<0.001) and Senegal (CVDRF 0.44 [0.40,0.60] vs HIV 0.40 [0.32,0.56], p<0.001).

When considering facility type for all countries, primary care had a significantly greater HIV than CVDRF derived readiness score (CVDRF, 0.20 [IQR 0.14,0.40] vs HIV 0.39 [0.28,0.50], p<0.001), as did secondary care (CVDRF 0.52 [0.38,0.66] vs HIV 0.60 [0.44,0.76], p<0.001). Publicly managed facilities scored significantly higher for HIV derived readiness than CVDRF (CVDRF, 0.20 [0.14,0.40] vs HIV 0.44 [0.32,0.58], p<0.001; while privately managed facilities had higher median CVDRF derived-readiness scores (CVDRF, 0.44 [0.24,0.58] vs HIV 0.40 [[0.28,0.60],p = 0.025). In both rural and urban facilities, HIV derived readiness scored higher than CVDRF (rural: CVDRF 0.22 [0.16,0.42] vs HIV 0.47 [0.28,0.59], p <0.001; urban: CVDRF 0.46 [0.34,0.62] vs HIV 0.48 [0.32,0.67], p = 0.004). Fig 2 shows a heatmap of this information disaggregated by readiness domain and stratified by country. For both CVDRF and HIV scores, information and staffing domains tend to be weakest, while equipment and diagnostics is stronger for CVDRF and amenities and medicines stronger for HIV.

When analyses were restricted to only facilities which reported readiness for either HIV or CVDRF (Table 2), considering all countries, derived readiness scores for both CVDRF (0.34 [IQR 0.18,0.52]) and HIV (0.68 [IQR 0.56,0.76]) were greater relative to scores when all facilities were considered; the derived readiness score for HIV remained greater than for CVDRF in this analysis (p < 0.001). Additionally, the relative increase in median score was greater for HIV (0.25) than CVDRF (0.06). This pattern of greater coverage score for HIV than for CVDRF care was similar in primary care, as was the greater increase in median score for HIV (0.22) compared

**Table 2. Readiness of facilities to provide Cardiovascular Disease Risk Factors (CVDRF) and HIV services.**

| | | | Facility-reported readiness | | | Derived readiness scores | | | Derived readiness scores in facilities with facility-reported readiness | | |
|---|---|---|---|---|---|---|---|---|---|---|---|
| | | All Facilities, n | CVDRF available, n (%) | HIV available, n (%) | CVDRF vs HIV Availability P-Value | CVDRF Score | HIV Score | CVDRF vs HIV P-Value | CVDRF Score | HIV Score | CVDRF vs HIV P-Value |
| Overall | All Facilities | 7522 | 6502.0 (86.4) | 2290.0 (30.4) | <0.001 | 0.28 (0.16,0.50) | 0.43 (0.32,0.60) | <0.001 | 0.34 (0.18,0.52) | 0.68 (0.56,0.76) | <0.001 |
| Country | Afghanistan | 142 | 118 (83.1) | 4 (2.8) | <0.001 | 0.52 (0.44,0.65) | 0.40 (0.32,0.47) | <0.001 | 0.54 (0.46,0.68) | 0.7 (0.64,0.75) | 0.885 |
| | Bangladesh | 1524 | 1168 (76.6) | 0 (0.0) | <0.001 | 0.16 (0.12,0.36) | 0.50 (0.28,0.50) | <0.001 | 0.2 (0.16,0.4) | NA | NA |
| | DRC | 1380 | 1196 (86.7) | 762 (55.2) | <0.001 | 0.36 (0.16,0.52) | 0.52 (0.32,0.68) | <0.001 | 0.4 (0.2,0.56) | 0.68 (0.56,0.8) | <0.001 |
| | Haiti | 1007 | 985 (97.8) | 158 (15.7) | <0.001 | 0.42 (0.24,0.58) | 0.32 (0.28,0.44) | <0.001 | 0.42 (0.24,0.58) | 0.74 (0.64,0.87) | <0.001 |
| | Malawi | 977 | 869 (88.9) | 615 (62.9) | <0.001 | 0.24 (0.16,0.42) | 0.53 (0.36,0.68) | <0.001 | 0.28 (0.18,0.44) | 0.61 (0.52,0.72) | <0.001 |
| | Nepal | 963 | 882 (91.6) | 61 (6.3) | <0.001 | 0.16 (0.12,0.25) | 0.32 (0.28,0.39) | <0.001 | 0.16 (0.12,0.28) | 0.57 (0.52,0.68) | <0.001 |
| | Senegal | 341 | 331 (97.1) | 84 (24.6) | <0.001 | 0.44 (0.40,0.60) | 0.40 (0.32,0.56) | <0.001 | 0.46 (0.4,0.62) | 0.7 (0.6,0.82) | <0.001 |
| | Tanzania | 1188 | 953 (80.2) | 606 (51.0) | <0.001 | 0.38 (0.18,0.60) | 0.52 (0.36,0.69) | <0.001 | 0.46 (0.24,0.66) | 0.68 (0.59,0.8) | <0.001 |
| Level of Care | Primary Care | 5634 | 4673 (82.9) | 1193 (21.2) | <0.001 | 0.20 (0.14,0.40) | 0.39 (0.28,0.50) | <0.001 | 0.24 (0.16,0.44) | 0.61 (0.52,0.72) | <0.001 |
| | Secondary Care or above | 1888 | 1829 (96.9) | 1097 (58.1) | <0.001 | 0.52 (0.38,0.66) | 0.60 (0.44,0.76) | <0.001 | 0.52 (0.4,0.66) | 0.72 (0.6,0.84) | <0.001 |
| Management | Private | 2703 | 2447 (90.5) | 809 (29.9) | <0.001 | 0.44 (0.25,0.58) | 0.40 (0.28,0.60) | 0.025 | 0.46 (0.3,0.6) | 0.68 (0.56,0.8) | <0.001 |
| | Public | 4819 | 4055 (84.1) | 1481 (30.7) | <0.001 | 0.20 (0.14,0.40) | 0.44 (0.32,0.58) | <0.001 | 0.24 (0.16,0.44) | 0.67 (0.56,0.76) | <0.001 |
| Urban-Rural* | Rural | 4281 | 3550 (82.9) | 1469 (34.3) | <0.001 | 0.22 (0.16,0.42) | 0.47 (0.28,0.59) | <0.001 | 0.26 (0.16,0.46) | 0.64 (0.56,0.76) | <0.001 |
| | Urban | 2278 | 2070 (90.9) | 760 (33.4) | <0.001 | 0.46 (0.34,0.62) | 0.48 (0.32,0.67) | 0.004 | 0.48 (0.38,0.64) | 0.72 (0.6,0.84) | <0.001 |

DRC–Democratic Republic of Congo. Facility-reported readiness of services, derived readiness scores for all facilities, and derived readiness scores for facilities reporting readiness to provide services are presented overall and by descriptive variables. Scores are presented by median (interquartile range). Maximum derived readiness score is 1. Detailed scores disaggregated by country are presented in S4,S5,S6 Appendices. *Nepal facilities excluded.

with CVDRF (0.04). No increase in median CVDRF score was seen for secondary care facilities, which is unsurprising as 96.9% of facilities reported readiness for CVDRF care.

When considering only those facilities which reported readiness, there was substantial variation in derived readiness scores between countries. In individual countries, derived readiness scores for CVDRF were generally lower than for HIV except in Afghanistan, where there was no evidence of a difference in medians (CVDRF 0.54 [IQR (0.46,0.68); HIV 0.70 [(0.64,0.75)], p = 0.885). Patterns of scores in individual countries were further nuanced when considering whether services were primary or secondary care, funded privately or publicly, or located in an urban or rural setting (Table 2). Fig 3 shows a heatmap of this information disaggregated further by readiness domain; while for CVDRF, weaknesses consistently remain in information, staffing and medicines, for HIV domains there is considerable improvement compared in facilities reporting readiness to provide HIV services. Availability of individual items used in

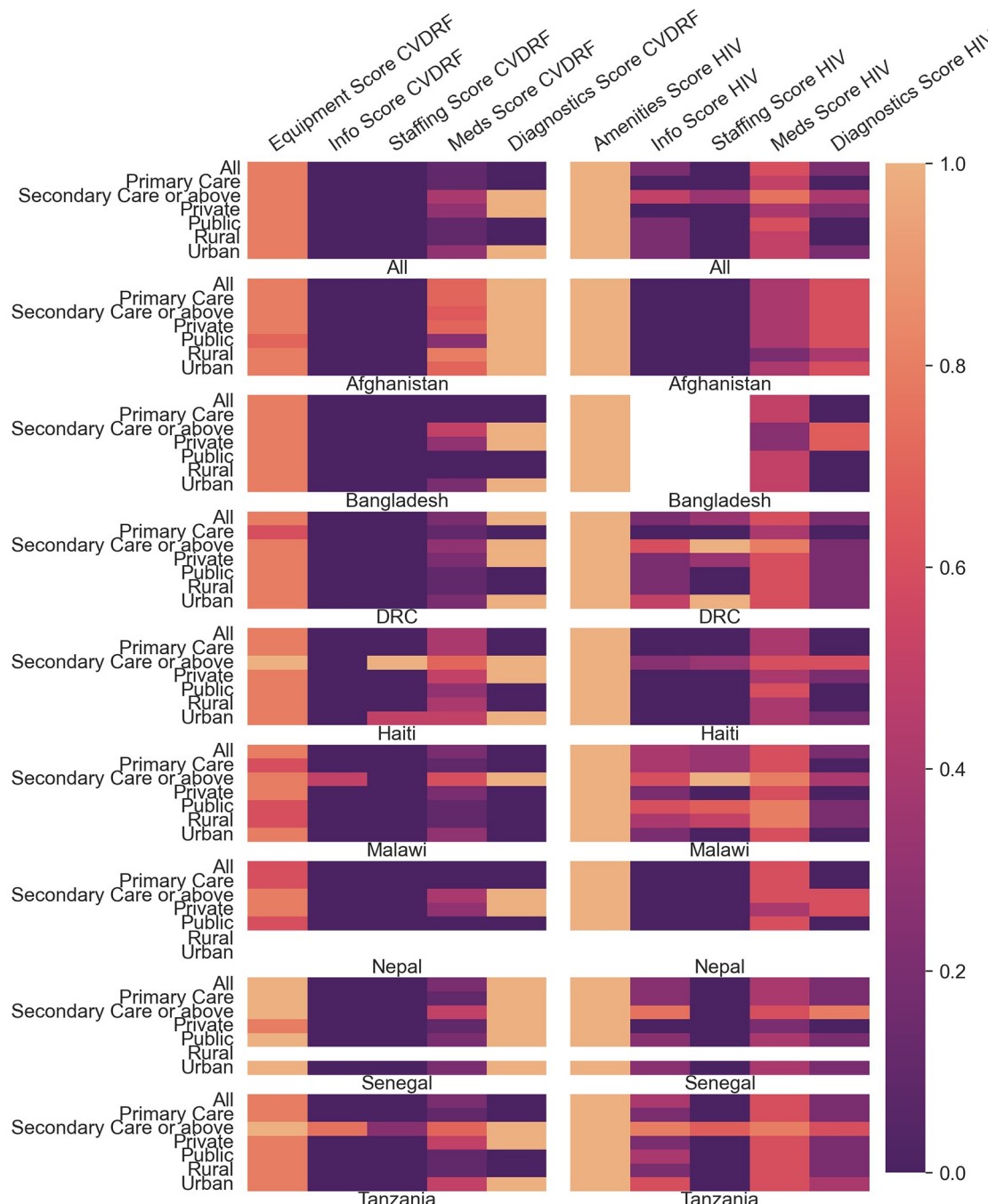

**Fig 2. Heatmap showing median facility derived readiness scores regardless of reported readiness to provide care for CVDRF or HIV.** CVDRF–Cardiovascular Disease Risk Factors; DRC–Democratic Republic of Congo. Scores reported for all and each individual country, by level of facility, funding, and geographical location (see **S5 Appendix** for data in table form) and disaggregated by readiness domain. Colorbar to the right of the figure represents median coverage score.

the calculation of derived readiness ranged from 96% of facilities possessing a stethoscope, to only 6% stocking statins (see S8 Appendix).

Considering all facilities regardless of reported readiness, multivariable analyses was conducted excluding Bangladesh and Nepal due to missing HIV availability and geographic

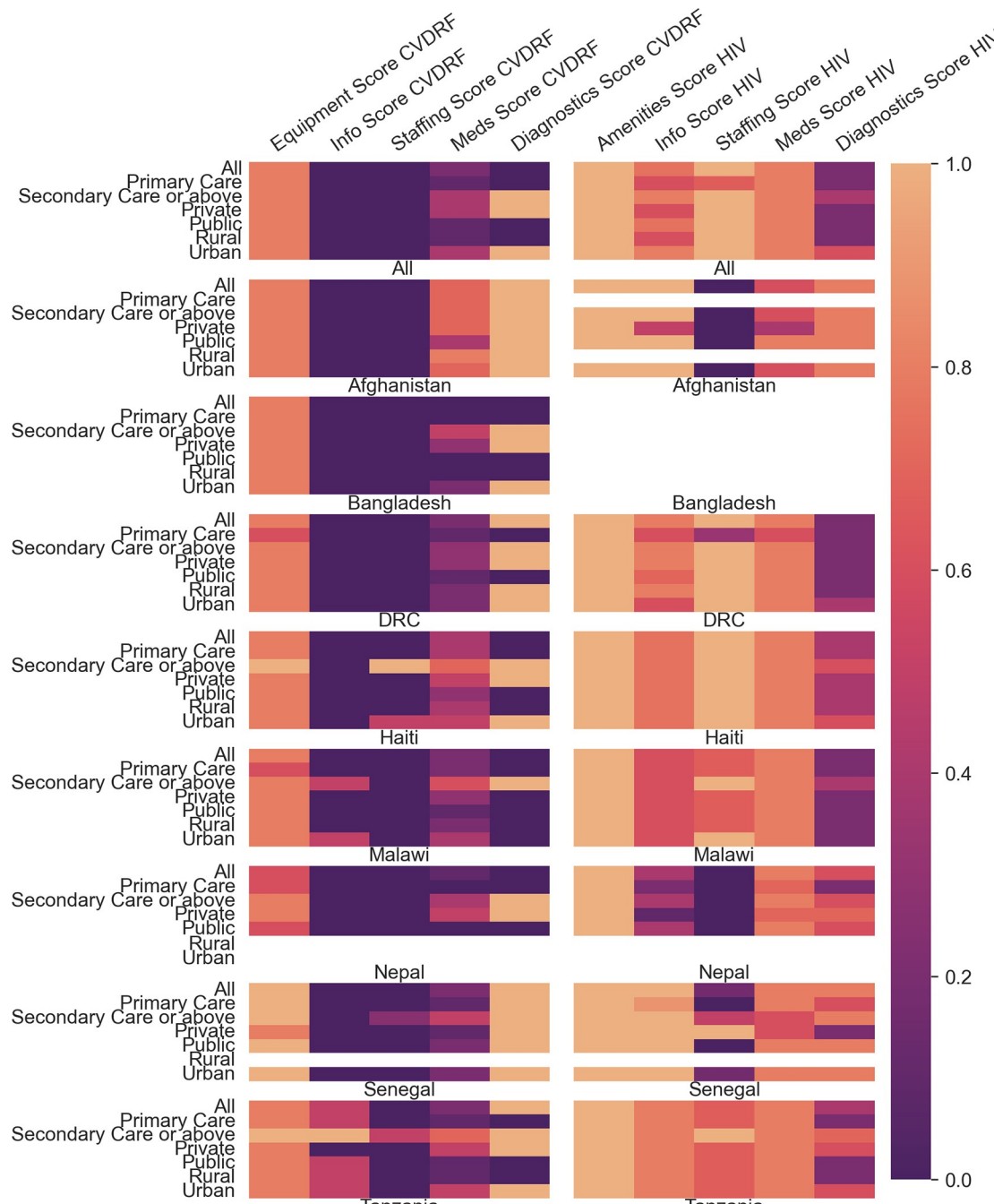

**Fig 3. Heatmap showing median derived readiness scores in facilities reporting readiness for that score.** CVDRF–Cardiovascular Disease Risk Factors; DRC–Democratic Republic of Congo. Scores reported for all and each individual country by level of facility, funding, and geographical location, restricting facilities to those which reported readiness for HIV or CVDRF (see S6 Appendix for data in table form). Colorbar to the right of the figure represents median readiness score.

setting data, respectively. Table 3 shows odds of greater CVDRF readiness were higher in urban areas, secondary care, and private facilities; odds of scores also vary by country. Addition of HIV derived readiness score in Model 2 increases the odds of higher CVD readiness. Odds of HIV readiness was greatest in urban areas, secondary care, and public facilities, and

**Table 3. Shows multivariable regression for Cardiovascular Disease Risk Factors (CVDRF) and HIV readiness scores.**

| | Model 1 | | | | | | Model 2 | | | | | |
|---|---|---|---|---|---|---|---|---|---|---|---|---|
| | CVDRF | | | HIV | | | CVDRF | | | HIV | | |
| | Odds Ratio | 95% CI | P-Value | Odds Ratio | 95% CI | P-Value | Odds Ratio | 95% CI | P-Value | Odds Ratio | 95% CI | P-Value |
| **DRC** | 1.000 (Ref) | | | 1.000 (Ref) | | | 1.000 (Ref) | | | 1.000 (Ref) | | |
| **Afghanistan** | 1.72 | 1.487–1.996 | <0.001 | 0.66 | 0.578–0.774 | <0.001 | 0.49 | 0.445–0.555 | <0.001 | 2.12 | 1.898–2.387 | <0.001 |
| **Haiti** | 2.39 | 2.246–2.560 | <0.001 | 1.06 | 0.977–1.149 | 0.163 | 0.61 | 0.568–0.666 | <0.001 | 2.47 | 2.335–2.630 | <0.001 |
| **Malawi** | 1.48 | 1.391–1.595 | <0.001 | 1.75 | 1.621–1.891 | <0.001 | 1.41 | 1.320–1.523 | <0.001 | 1.18 | 1.113–1.262 | <0.001 |
| **Senegal** | 1.83 | 1.680–2.012 | <0.001 | 1.23 | 1.096–1.397 | 0.001 | 0.86 | 0.776–0.964 | 0.008 | 1.76 | 1.626–1.904 | <0.001 |
| **Tanzania** | 1.76 | 1.650–1.879 | <0.001 | 1.67 | 1.557–1.797 | <0.001 | 1.18 | 1.112–1.271 | <0.001 | 1.44 | 1.357–1.527 | <0.001 |
| **Rural** | 1.000 (Ref) | | | 1.000 (Ref) | | | 1.000 (Ref) | | | 1.000 (Ref) | | |
| **Urban** | 1.28 | 1.221–1.347 | <0.001 | 1.08 | 1.023–1.143 | 0.006 | 0.91 | 0.868–0.965 | 0.001 | 1.21 | 1.165–1.274 | <0.001 |
| **Primary Care** | 1.000 (Ref) | | | 1.000 (Ref) | | | 1.000 (Ref) | | | 1.000 (Ref) | | |
| **Secondary Care or above** | 2.62 | 2.484–2.765 | <0.001 | 2.96 | 2.784–3.152 | <0.001 | 1.65 | 1.551–1.754 | <0.001 | 1.65 | 1.567–1.740 | <0.001 |
| **Public** | 1.000 (Ref) | | | 1.000 (Ref) | | | 1.000 (Ref) | | | 1.000 (Ref) | | |
| **Private** | 1.25 | 1.197–1.307 | <0.001 | 0.71 | 0.679–0.751 | <0.001 | 0.58 | 0.562–0.618 | <0.001 | 1.44 | 1.390–1.505 | <0.001 |
| **HIV Score** | | - | | | - | | 13.8 | (12.1–2,15. | <0.001 | | - | |
| **CVDRF Score** | | - | | | - | | | - | | 6.23 | 5.675–6.841 | <0.001 |

DRC–Democratic Republic of Congo. Model 1 includes country, rural or urban location, facility level, and funding. Model 2 includes facility HIV readiness score for CVDRF and CVDRF readiness score for HIV. Results are Odds Ratio (95% CI), P-Value.

also varies by country. Model 2 shows that HIV readiness score was also greater in facilities with higher CVDRF scores.

## Discussion

In this multi country, multi facility secondary analysis of the Demographic and Health Surveys' SPA surveys, we show that many facilities are not ready to manage CVDRFs, especially due to the lack of key medicines and human resources. Most facilities had derived readiness scores for CVDRF below the WHO target of 80% for NCDs, of which the vast majority self-reported readiness for CVDRF services [36]. Although facility-reported readiness was greater than derived readiness scores, median readiness score for CVDRF substantially increased when analyses were restricted to facilities which reported providing services for CVDRF. Our results also show a striking over-estimation of, faciity-reported, readiness to provide CVD care and underestimation of readiness to provide HIV care. This suggests that facility reported readiness should not be used as a metric to assess actual readiness.

Low objectively assessed readiness of primary care to provide CVDRF services is of particular concern given that CVDRF interventions such as detailed in the WHO PEN guidelines are intended for delivery primarily through primary care services [27,37]. Given that facility readiness to provide services is a pre-requisit for care being delivered, results of low readiness are commensurate with findings of poor transition through the cascades of care for diabetes and hypertension in the countries included in this study [5,6]. They are also reflective of findings of WHO NCD Country Profiles [38] which have shown substantial variation in implementation of national targets and availability of key medicines, equipment and guidelines at facilities. In combination, the findings bring into question the ability of the global community to achieve targets of the UN High Level Commmission on NCDs, to reduce mortality from the four major NCDs by 25% between 2010 and 2025, or SDG 3.4 to reduce premature mortality from NCDs by one third by 2030 [3,36].

Despite increased and sustained global attention for HIV relative to CVDRF, we found that derived coverage for HIV care, although higher than for CVDRF care, was also low, even in the countries with a relatively high HIV burden such as Tanzania and Malawi. Our results show this is driven by lack of diagnostic testing in Tanzania, and by lack of staffing and information systems in Malawi. These weaknesses are also noted in the respective SPA country report [39]. The complexity of delivering HIV care may mean that even greater investments are needed to ensure service readiness [20]. Nevertheless, higher HIV than CVDRF coverage scores almost certainly result from historically greater resources and policy attention [40].

Patterns of derived readiness scores for HIV were similar to CVDRF care, although whereas CVDRF readiness scores were greater in private facilities, for HIV public facilities were more ready. This may reflect the investment that governments have made into improving services for HIV as a result of the MDGs, while CVDRFs have previously been seen as diseases of the wealthy and so have received more attention in the private sector [41]. Interestingly, our multivariable regression showed that HIV and CVDRF readiness scores were closely associated. While there may be unmeasured factors that facilitate service readiness for multiple conditions, it is possible that developed HIV services directly support the readiness for CVDRF services. This relationship is also noted in studies of care processes for CVDRF and HIV, where treatment and control of HIV is associated with greater awareness and treatment of hypertension [42]. Both ART programs and CVDRF care require long term monitoring and management of conditions; this necessitates readiness not only in terms of equipment and medicines, but also clear care pathways, health records, and patient recall systems. Converting chronic ART programs into chronic disease management platforms may represent a route to better integrating HIV programmes into health systems, and to providing the longitudinal infrastructure to deliver CVDRF care to patients over many years [4,15]. Together these findings suggest that integration of services may represent a roadmap for improving CVDRF readiness, with opportunities to capitalise on the multidisciplinary teams and the infrastructure needed to manage these chronic diseases [16]. Building on this idea, the Ideal Clinical program in South Africa plans for more integration of services [43].

Low health service readiness is likely an important factor in explaining why cascades of care show that individual access to effective treatments for CVDRFs is low [4,7,11]. For example, despite our finding that (self-reported) services are provided at the majority of facilities, vanishingly few actually had a statin available, which corresponds closely to Flood et al.'s [4] finding that under 5% of individuals with diabetes were appropriately managed, driven by only 6.3% having access to cholesterol lowering treatment. But our findings of facility readiness scores do not entirely match health system performance as measured in the cascades. For example, whilst studies have shown that around 17% of people with hypertension in Bangladesh had their blood pressure well-controlled [5]–which is relatively high, compared to other countries—our results show median coverage score of just 0.16. Conversely, 2.2% of people with hypertension in Tanzania had this controlled in analyses of the cascades of care, but we found median CVDRF readiness score in Tanzania was relatively high at 0.38. These results concur with Davies et al's findings that although facility readiness is a necessary component for ensuring transit through the cascades, it is not sufficient [11]. Access to quality care leading to adequate treatment coverage or disease control goes beyond the availability of ready health services; other elements of access to quality care are required to match service provision with outcomes. These include cultural norms, geo-temporal access, affordability of services, the quality of the services provided, and individual and societal behavioural factors that affect care uptake, especially for chronic diseases [44,45].

Facility readiness may vary due to country level factors such as national policy, Direct Assistance for Health (DAH) funding, or national burdens of diseases, although the relationships

between these factors and readiness may vary by country, given the multiple interacting factors which influence health systems performance. For example, 1% of DAH in Malawi was spent on NCD care in the year the survey was conducted [40], the highest proportion of all countries investigated in this study, where median CVDRF readiness scores were 0.24. In contrast, 36% of DAH spending in Malawi was allocated to HIV and median scores were 0.53. Conversely, in Tanzania 0.1% of DAH was allocated to NCDs and the median CVDRF score was 0.38, whereas for HIV allocation was 48% and the median score was 0.52.(11)Our measures of facility readiness should also be affected by disease epidemiology, however, the prevalence of HIV or CVDRF does not consistently map onto the differences in service readiness that we found. For example, in Malawi, DRC and Nepal, readiness scores for HIV were 0.53, 0.52 and 0.32 respectively, where HIV deaths per 1000 deaths were 156.41, 18.53 and 6.73 respectively in the years that the surveys were done (see S9 Appendix for tabular data) [1]. Similarly, in Malawi, DRC and Nepal, CVDRF median coverage scores were 0.24, 0.36 and 0.16, while CVD deaths per 1000 deaths were 87, 101 and 140 [1]. We compare scores to deaths rather than disease prevalence as deaths are more likely to reflect unmet healthcare needs than prevalence of conditions where monitoring of disease prevalence is not regularly used.

Government policies and their implementation is also likely to influence facility readiness, but these alone are unlikely to lead to improved readiness. Findings from NCD policy readiness scores reported by Allen et al [46] did not reflect CVD facility readiness scores that we saw in our study; for example the policy readiness score reported for Bangladesh was 38.9% and DRC 34.2%, where our median CVDRF readiness score in Bangladesh was 0.16 and in DRC was 0.36.

Our results compare to previous studies which have used composite scores from SPA or SARA to assess readiness to provide care for other conditions [24,47,48]. However, our study updates the findings [49,50] by including surveys from additional countries, updating the timeframe, and comparing with HIV service readiness. Like these previous studies, we show that readiness for CVDRF remains low with small differences between facilities stating that they offer services and those that do not; however, our data shows that for HIV care, there are larger increases in readiness between facilities stating that they offer HIV services and those that do not. We additionally show the clear associations between readiness for CVDRF and HIV.

## Limitations

Our study has several limitations. This study is not able to directly link readiness to receipt of high quality care, and there may be unmeasured variation in the ability of different countries to convert the necessary but not sufficient readiness into high quality care.

There was only data available for 8 LMICs with HIV burden varying substantially between countries, and findings may not be applicable to other settings. We included only one survey from each country selecting the most recent survey to ensure results are as pertinent to current heath service readiness as possible, including surveys up to 10 years old. Despite requesting data from the SARA surveys from WHO, these were not made available. If SARA data were available, at least 7 more studies would meet our inclusion criteria. Cross-sectional survey designs may not represent readiness well as components may be missing on the day only, or stocks of components such as medicines may represent underuse of that component. Readiness scores present a summary only, and some components could be considered more important than others to delivering care. For example, a facility with no access to medications cannot offer many treatments even if all equipment is present. Conversely, not all components are required for a facility to provide an effective service; for example, a dispensary may be located near a health facility with all essential medications available to facility clients, but the

facility would still score poorly for readiness if the facility does not stock essential medicines. However, although considered an essential means of assessing service readiness, the SPA surveys did not allow us to provide such a nuanced analysis, additionally the issues we have described would likely be similar for CVDRF and HIV readiness. We used the question on self-reported readiness to provide NCD care as a marker of whether they provided CVDRF care, whereas facilities may have been responding positively to provision of care for respiratory diseases such as Chronic Obstructive Pulmonary Disorder rather than CVDRF–this may have led to a falsely high number of facilities subjectively reporting providing CVDRF services.

The findings may not be transferable to all LMIC contexts; for example, only 2 low-income countries were included and no upper-middle income countries. These surveys took place between 2014–18 and so may no longer reflect current readiness or practice in HIV and CVDRF care, although they still allow comparison between HIV and CVDRF readiness. Facilities were included from 8 different surveys and some variation exists between them particularly in facility selection, despite rigorous standardisation of the methodology. For example, Afghanistan's SPA focused on private primary care facilities and all secondary care facilities. However, other than Afghanistan, surveys were nationally representative, and Afghanistan was included as we belive it is still able to contribute valuable information of comparative facility readiness between HIV and CVDRF. Data were not weighted by sampling frames, and instead stratified analyses presented to understand the importance of different facility types. This is similar to the approach taken in other international comparisons of cross-sectional survey data [4,5]. We also excluded facilities which did not respond to the survey; however 96.5% of facilities completed the survey minimising issues of selection bias. Data on missingness are included in S10 Appendix.

We combined the SARA manual's indices for diabetes and CVD. These components represented most requirements for managing CVDRFs, but this score didn't exclusively focus on CVDRF nor considered all risk factors [20]. The SARA index for CVDRF components has several key gaps in assessing CVDRF care compared to the PEN guidelines requirements, in particular the absence of statins for managing CVDRF conditions. Our sensitivity analysis using PEN components did not suggest inclusion additional PEN components changed the results substantively. Finally, the SPA contained small variations between surveys, in particular in analgesics and in anti-fungal drugs. This led to the exclusion of IV antifungals as a component of HIV care, as no survey which we utilised collected this item, but does not substantially affect the comparisons and the denominators used to assess readiness were consistent across countries.

## Conclusions

This study, which used a large, international dataset of nationally representative surveys with a stable inventory of questions over the study period, showed that the vast majority of health facilities surveyed in LMICs using the SPA tool are unprepared to deliver CVDRF services. It also indicated that sustained investment in improving care for chronic diseases, such as HIV, may be associated with improved readiness, but that even with this investment readiness remains suboptimal.

With increased focus on NCDs as part of the SDG agenda and through initiatives such as the Diabetes Compact [51,52], the WHO's "vision of reducing the risk of diabetes, and ensuring that all people who are diagnosed with diabetes have access to equitable, comprehensive, affordable and quality treatment and care," there may be substantially greater opportunities for policy makers to improve CVDRF care. Our findings provide policy makers, funders, and researchers with evidence of where there are gaps in service provision which need to be filled to enable achievement of current global health goals.

## Supporting information

**S1 Appendix. Data dictionary.** Detailed explanation and definition of all variables used in the analysis.
(XLSX)

**S2 Appendix. Variables by source and country.** Variables used to derive Health Service Readiness scores for HIV and CVDRF shown by domain.
(XLSX)

**S3 Appendix. Score comparisons.** Comparison between CVDRF score derived using the SARA manual or the PEN guidelines.
(DOCX)

**S4 Appendix. Facility reported readiness to provide services.**
(XLSX)

**S5 Appendix. Median all-facility readiness scores.** Facility readiness to provide CVDRF or HIV care, regardless of stated readiness of services to provide care for CVDRF or HIV presented overall and stratified by level of care, funding, and geographical location. Scores are presented as median (IQR).
(XLSX)

**S6 Appendix. Median readiness in facilities reporting readiness.** Facility readiness to provide CVDRF or HIV care for facilities stating readiness of services to provide care for CVDRF or HIV presented overall and stratified by level of care, funding, and geographical location. Scores are presented as median (IQR).
(XLSX)

**S7 Appendix. Stated readiness for HIV and CVDRF crosstabbed.** Number (%) of facilities which stated readiness to provide none, either or both CVDRF and HIV care.
(XLSX)

**S8 Appendix. Individual variable results.** Availability of individual components of care used in the construction of the CVDRF or HIV Health Service Readiness scores.
(XLSX)

**S9 Appendix. Disease specific deaths and service readiness.**
(XLSX)

**S10 Appendix. Missing facilities by country and type.** Number of facilities that did not complete the survey, by country and by facility type. 3.5% of facilities did not complete surveys.
(XLSX)

**S11 Appendix. Sensitivity analysis with scores unweighted by domain.** All results and analyses throughout the paper are repeated without weighting individual domains of CVDRF and HIV equally i.e. all items are weighted equally.
(XLSX)

**S12 Appendix. Sensitivity analysis using second-most recent national survey where multiple surveys available.** All results and analyses throughout the paper are repeated using the penultimate survey available for Bangladesh (2014), Senegal (2018) and Haiti (2013), and the one available for all other countries as reported in the main paper.
(XLSX)

## Acknowledgments

PG is a Chan Zuckerberg Biohub investigator.

## Author Contributions

**Conceptualization:** Justine Davies.

**Data curation:** Neil Cockburn.

**Formal analysis:** Neil Cockburn.

**Investigation:** Neil Cockburn.

**Methodology:** Neil Cockburn, David Flood, Jacqueline A. Seiglie, Jennifer Manne-Goehler, Justine Davies.

**Project administration:** Neil Cockburn.

**Supervision:** Justine Davies.

**Visualization:** Neil Cockburn.

**Writing – original draft:** Neil Cockburn, Krishna Aryal, Justine Davies.

**Writing – review & editing:** Neil Cockburn, David Flood, Jacqueline A. Seiglie, Jennifer Manne-Goehler, Khem Karki, Albertino Damasceno, Rifat Atun, Sebastian Vollmer, Till Bärnighausen, Pascal Geldsetzer, Mary Mayige, Lisa Hirschhorn, Justine Davies.

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
