## [Decision Letter · Decision Letter 0]

11 May 2023

PGPH-D-23-00299

Health service readiness to provide care for HIV and cardiovascular disease risk factors in low- and middle-income countries: a comparative analysis of data from Service Provision Assessments

Dear Dr. Cockburn,

Thank you for submitting your manuscript to PLOS Global Public Health. After careful consideration, we feel that it has merit but does not fully meet PLOS Global Public Health’s publication criteria as it currently stands. Therefore, we invite you to submit a revised version of the manuscript that addresses the points raised during the review process.

We look forward to receiving your revised manuscript.

Kind regards,

Nasheeta Peer

Academic Editor

Journal Requirements:

1. We have noticed that you have uploaded Supporting Information files, but you have not included a list of legends. Please add a full list of legends for your Supporting Information files after the references list. 

Additional Editor Comments (if provided):

Overall:

The key overarching factors/categories influencing optimal healthcare provision include 1) governance i.e., government policies/strategies/guidelines, 2) healthcare system factors, 3) healthcare provider factors and 4) patient factors. These are multi-level, multiple and complex factors that interact and influence each other. Therefore, evaluating a single category is not necessarily a direct reflection of the ability to provide optimal care for a particular condition. The authors have failed to acknowledge these complexities in their paper.

Have SARA or SPA been validated? That is, found to be useful indicators of care for CVDRFs and HIV? What are the other factors that influence care that have not been considered?

Line 68: Please introduce the acronym LMICs here and not in line 75.

Line 69: please be consistent with your use of introduced acronyms – CVD.

Please pay close attention to acronyms throughout the paper e.g., line 85 – introduce acronym MDG before using it.

Lines 117-118: state clearly that this study focused on the 2012 surveys that were conducted in…list the countries.

Lines 134-140: The outcomes should be incorporated into the aims (currently vague) in lines 105-106. Do not present this section in point format.

It is unclear if the authors used SPA or SPA, and how does PEN fit in? Please explain clearly.

Line 141: What is considered a high score vs. a moderate or low score for the various outcome measures? These need to be clearly elucidated for the reader.

Line 201: The following is incorrect: ‘All variables are described as mean (%) when categorical…’ – means are not reported for categorical data.

Were the data skewed?

Line 216: How many low-income and how many middle-income countries were included? Were there any differences between these 2 categories?

What do the overall scores indicate in terms of readiness for provision of care in these 8 countries? Please provide the overall context before reporting on the comparisons.

Was coverage good or poor? Please describe the study findings using such phrases rather than simply presenting the data.

Lines 226-228: This belongs in the Methods.

All tables should be standalone with acronyms explained in the text or footnotes – e.g., DRC, CVD/CVDRF

The results are difficult to follow. Please consider presenting the overall key values/data in the text and summarising those for each country.

Line 262: I could not find Fig 21. Please ensure correct labelling of all tables and figures.

The Discussion is difficult to follow. The authors could perhaps make the text easier to follow by clearly describing what each outcome being discussed represents and summarising the key findings in the first paragraph.

Lines 303-304: What were the coverage scores?

Line 306: ‘21% greater’ than what? This sentence is unclear.

Lines 319-321: Can the authors provide a reason/s for the low HIV coverage in Tanzania and Malawi? This would be of interest to the readers.

Line 360-361: please rephrase – what was controlled?

Lien 363: What then is necessary? And so, what is the relevance of this study? And why is there a lack of concordance?

Line 366: What is inconsistent? The authors need to improve their writing style and ensure clarity in their written text.

Line 374: ‘differences in facility readiness’ – what is the comparison here?

Line 379: Please include the reference for the prevalence. Are you comparing like with like? Please expand/explain. How reliable are the data i.e., death certification in these countries?

Lines 382-383: Policies do not always translate into action i.e., implementation, and implementation is not necessarily effective, nor does it directly address what is needed. Please rework this section.

Lines 387-399: Some of this paragraph belongs at the beginning of the Discussion where the authors summarise their study findings.

Line 423: please correct spelling. What respiratory diseases care are related to CVDRFs?

Lines 429 and 447 contradict each other in terms of representativeness. Please amend/clarify accordingly.

Lines 452-453: Briefly clarify/explain the Diabetes Compact (one sentence or phrase)

Line 456: The conclusions should be an overall conclusion without referring to a single component, in this instance statins. Please rephrase. Further, is there testing of lipid profiles to warrant the stocking of statins? In low-income countries, ensuring access to adequate diabetes medicines is difficult enough, let alone focusing on statins. Was there sufficient access to diabetes and hypertension medications in the study countries?

Reviewers' comments:

Reviewer's Responses to Questions

**Comments to the Author**

1. Does this manuscript meet PLOS Global Public Health’s publication criteria? Is the manuscript technically sound, and do the data support the conclusions? The manuscript must describe methodologically and ethically rigorous research with conclusions that are appropriately drawn based on the data presented.

Reviewer #1: Partly

Reviewer #2: Yes

Reviewer #3: Yes

Reviewer #4: Yes

Reviewer #5: Yes

2. Has the statistical analysis been performed appropriately and rigorously?

Reviewer #1: Yes

Reviewer #2: Yes

Reviewer #3: Yes

Reviewer #4: No

Reviewer #5: Yes

3. Have the authors made all data underlying the findings in their manuscript fully available (please refer to the Data Availability Statement at the start of the manuscript PDF file)?

Reviewer #1: Yes

Reviewer #2: Yes

Reviewer #3: Yes

Reviewer #4: Yes

Reviewer #5: Yes

4. Is the manuscript presented in an intelligible fashion and written in standard English?

Reviewer #1: Yes

Reviewer #2: Yes

Reviewer #3: Yes

Reviewer #4: Yes

Reviewer #5: Yes

5. Review Comments to the Author

Reviewer #1: I commend the purpose and premise of this analysis, however, although this involves a large data set, I don’t think this summary data answers the research purpose well, and the data has limited translatability.

As outlined, the investment in HIV clinical research, HIV healthcare implementation and innovation has revolutionised the management of HIV enabling people living with HIV to live healthy, normal lives. Such global investment has not been observed for many other neglected communicable and non-communicable diseases (NCDs).

This research highlights the challenge of enabling better healthcare outcomes for non-communicable diseases (specifically cardiovascular disease and diabetes) associated with high morbidity and mortality, considering how we can leverage the achievements and integrate NCD/CVS care within HIV care in LMICs.

The article is written well, with clarity and with succinct descriptions.

My major issues are related to the methodology and its application to answer the challenge posed in the introduction:

Cross-sectional data is taken from SPA surveys after 2012 (2014-2018) from eight countries and 7911 facilities.

The SPA tool is based on the WHO SARA tool, and includes additional questions to evaluate the quality of health services provided at health facilities. SPA assesses both the availability and quality of health services provided, including family planning, maternal and child health, HIV/AIDS, and other health services.

Overall, both SARA and SPA are useful instruments for assessing the readiness of health facilities to provide essential health services.

1) However, there are a few critical factors to consider when using these tools. For example, they rely heavily on the accuracy and completeness of data collected from health facilities, which can be challenging to obtain in some settings. Additionally, the indicators used to assess readiness may not be suitable for all settings and may need to be adapted to local contexts. Hence there is a need for caution, and it requires a combination with other data sources to provide a comprehensive assessment of health service delivery.

2) There is a large variation in the timing of data/results from countries (data from surveys completed in 2014 – 2018). Need to take into account the changes in services that may have occurred since surveys were completed, e.g >5 years ago.

3) From the tables – there is a significant variation in the ‘% of total surveyed’ across countries, for all three categories – Level of Care (primary/secondary), Management (private/public) and Geographic location (rural/urban)

Hence, interpretation of this data is limited due to the highly varied implementation of the WHO survey to assess readiness and service provision across primary and secondary care facilities across regions and across countries.

There is limited discussion about the potential bias related to why and which services would routinely and completely complete these surveys.

Given the vast differences/variations of contexts and health systems for regions, particularly across eight diverse countries, the translation of this data is limited.

Translation of this data should be aligned with more robust service data that tailor surveys/investigations to the individual country and region contexts and health systems.

The tables clearly highlight this variation, although there is little content that highlights the vast differences in regions' contexts and health systems.

The paper’s point about leveraging HIV healthcare system provision to manage CVS disease and diabetes is an important topic – although this data outlines a lack of preparedness to deliver CVDRF services and overall improved readiness to provide HIV services, much of this data may be outdated and lack representation. Additionally, it doesn’t assess how to bridge HIV healthcare system gaps and services for NCDs; and the ability to leverage HIV services.

The heatmap was not a useful representation and comes across as quite cluttered.

I think this large dataset is useful to interpret and highlight but the limitations regarding its meaning and translatability require to be discussed more robustly and clearly.

Reviewer #2: Title: Health service readiness to provide care for HIV and cardiovascular disease risk

factors in low- and middle-income countries: a comparative analysis of data from

Service Provision Assessments

Title(s)

Suggestion to make the running title read ‘Health service readiness to provide care for HIV and cardiovascular disease risk factors in low- and middle-income countries’

Abstract

Abstract is simplified and well written

Introduction

Background to study is adequately provided, and objectives are clearly stated

Materials and Methods

- Line 190 onwards: Should it not read?’ Scores were created for six (instead of five stated here) domains …. categorized as amenities, equipment, diagnostics, information, medicines, and staffing, …..rather?

Results

- Results are well represented with appropriate tables, figures and links to other data.

Discussions

- With the limitations in mind, the results are adequately discussed, and it makes reference to similar and other relevant studies.

General Comments/Summary

1. This is an important study. With its well acknowledged limitations, it would positively influence policies of Governments and Health facilities (in LMIC's), to improve CVD, CVDRF and HIV patient management and outcome.

Reviewer #3: Overall comment

• It is a useful study. However, the main concern is old data from few countries. Another major concern is the lack of clarity as to how the program managers will use this information to improve the facility's readiness. In its current form, it is a highly theoretical academic exercise.

Specific comments

Introduction

• The rationale for comparing HIV and CVD is not clear. These services are very different, and HIV programs have existed for much longer.

Methods

• Explain the key variables included in the five domains mentioned in Figure 2.

• Define what is included in CVDRF services.

Results

• Table 2 – N for facilities included in the “coverage score” and facilities included in the analysis for “Coverage scores in facilities with self-reported” should be mentioned before the mean score.

• Was the availability of drugs/diagnostics included in the scores for the private sector? This may not be relevant as most private doctors only give prescriptions to buy drugs from pharmacies/labs.

• Present more specific information about the gaps in each of the domains (at least a few) to make the data useful for program managers.

Discussion

• Due to the complexity and length of data collection tools, these survey tools may lead to poor-quality data in low-resource settings. Authors should reflect on how these surveys can be made more useful and practical based on their analysis. Are there any specific suggestions about changing/modifying some of the questions to make them more useful for analysis and interpretation?

• The authors have highlighted the disconnect between scores and treatment coverage/control in the discussion. It further raises the question about the usefulness of these surveys. It is important to reflect if the questions truly capture what is useful. Were there variables which were not useful and could be removed? Were there variables which could not be appropriately interpreted? For example, the issue of adequacy of medications which will depend on the patient load, may not answer the issue of adequacy of medications. If the information systems are weak, this information may not be available, hence limiting the usefulness of drug data.

• Limitations – Data for few countries is old, and facilities might have improved. Mention the countries for which data is more than 5 yr old.

• Recommendations are too generic. Make it specific by domain or country as relevant.

Reviewer #4: On whether the statistical analysis has been performed appropriately and rigorously;

The methodology has not factored in the varying disease burden for both HIV and NCD for the different countries, nor adjusted for the same in the statistical analysis.

In analyzing the scores, whether primary, secondary or other; rural versus urban, private versus public, the author is not clear on whether these facilities are matched based on a clear criteria for comparison across the different countries

Beyond restricting analysis to only facilities with self-reported service, the author could consider restricting to those with coverage scores as well in looking at access and management of NCDs to further support their conclusion.

Reviewer #5: Over the years communicable diseases have been prioritised over non-communicable diseases, the leading cause of morbidity and mortality worldwide especially, in low and middle income countries, making the research topic critical to global health. Overall, I would recommend this paper for publication with a minor revision.

The author gives a decent description of the research method. However I would recommend highlighting evidence to support the effectiveness and suitablity of the adopted method to the research topic. The limitations of the methods, I believe is worth mentioning here. Which of the services that inform readiness are operational or only available? Are there any bias?

Great use of heatmaps in results presentation. Generally good comparisons drawn in relation to public/private and rural/urban facilities.

In concluding, the author backs a statement (not supported by evidence) with cost-effectiveness of managing CVDRFs which could be misleading. Is this with respect to HIV here? Is this just a thought? This should be clear as even not all CVD interventions are that cost effective.

6. PLOS authors have the option to publish the peer review history of their article (what does this mean?). If published, this will include your full peer review and any attached files.

**Do you want your identity to be public for this peer review?** For information about this choice, including consent withdrawal, please see our Privacy Policy.

Reviewer #1: No

Reviewer #2: **Yes: **Eric NY Nyarko

Reviewer #3: No

Reviewer #4: No

Reviewer #5: **Yes: **Patriot Ofori-Aning

---

## [Decision Letter · Decision Letter 1]

22 Aug 2023

Health service readiness to provide care for HIV and cardiovascular disease risk factors in low- and middle-income countries

PGPH-D-23-00299R1

Dear Dr Cockburn,

We are pleased to inform you that your manuscript 'Health service readiness to provide care for HIV and cardiovascular disease risk factors in low- and middle-income countries' has been provisionally accepted for publication in PLOS Global Public Health.

Best regards,

Nasheeta Peer

Academic Editor

Reviewer Comments (if any, and for reference):

Reviewer's Responses to Questions

**Comments to the Author**

1. If the authors have adequately addressed your comments raised in a previous round of review and you feel that this manuscript is now acceptable for publication, you may indicate that here to bypass the “Comments to the Author” section, enter your conflict of interest statement in the “Confidential to Editor” section, and submit your "Accept" recommendation.

Reviewer #1: All comments have been addressed

Reviewer #2: All comments have been addressed

Reviewer #3: All comments have been addressed

2. Does this manuscript meet PLOS Global Public Health’s publication criteria? Is the manuscript technically sound, and do the data support the conclusions? The manuscript must describe methodologically and ethically rigorous research with conclusions that are appropriately drawn based on the data presented.

Reviewer #1: Yes

Reviewer #2: Yes

Reviewer #3: Yes

3. Has the statistical analysis been performed appropriately and rigorously?

Reviewer #1: Yes

Reviewer #2: I don't know

Reviewer #3: Yes

4. Have the authors made all data underlying the findings in their manuscript fully available (please refer to the Data Availability Statement at the start of the manuscript PDF file)?

Reviewer #1: Yes

Reviewer #2: Yes

Reviewer #3: Yes

5. Is the manuscript presented in an intelligible fashion and written in standard English?

Reviewer #1: Yes

Reviewer #2: Yes

Reviewer #3: Yes

6. Review Comments to the Author

Reviewer #1: The authors have responded to the reviewer’s comments and queries well.

I am happy to recommend this article for publication.

I suggest that the authors cut the text to enable a more concise read with a short table summarising the research tools, the results and discussion and limitations. This would also help to highlight this work in the general or tailored media.

I have a challenge with how translatable this research is given the methodology (lack of a robust research tool applied over a large geographical area), but it has an overall purpose to understand a wider picture and the need to strengthen CVS services and the management of NCDs globally; and in understanding the limitations and lack of infrastructure, preparedness/readiness and investment related to this challenge in different regions.

Reviewer #2: All comments have been addressed

Reviewer #3: (No Response)

7. PLOS authors have the option to publish the peer review history of their article (what does this mean?). If published, this will include your full peer review and any attached files.

**Do you want your identity to be public for this peer review?** For information about this choice, including consent withdrawal, please see our Privacy Policy.

Reviewer #1: No

Reviewer #2: No

Reviewer #3: No
